# A wall-time minimizing parallelization strategy for approximate Bayesian computation

**Emad Alamoudi**[1], **Felipe Reck**[1], **Nils Bundgaard**[2], **Frederik Graw**[2,3,4], **Lutz Brusch**[5], **Jan Hasenauer**[1,6,7]*, **Yannik Schälte**[1,6,7]

**1** Life and Medical Sciences Institute, University of Bonn, Bonn, Germany, **2** BioQuant—Center for Quantitative Biology, Heidelberg University, Heidelberg, Germany, **3** Interdisciplinary Center for Scientific Computing, Heidelberg University, Heidelberg, Germany, **4** Department of Medicine 5, Friedrich-Alexander-University Erlangen-Nürnberg, Erlangen, Germany, **5** Center of Information Services and High Performance Computing (ZIH), Technische Universität Dresden, Dresden, Germany, **6** Helmholtz Zentrum München, Institute of Computational Biology, Neuherberg, Germany, **7** Center for Mathematics, Technische Universität München, Garching, Germany

☯ These authors contributed equally to this work.
* jan.hasenauer@uni-bonn.de

**Data Availability Statement:** The code underlying this study can be found at https://github.com/EmadAlamoudi/Lookahead_study. A snapshot of

## Abstract

Approximate Bayesian Computation (ABC) is a widely applicable and popular approach to estimating unknown parameters of mechanistic models. As ABC analyses are computationally expensive, parallelization on high-performance infrastructure is often necessary. However, the existing parallelization strategies leave computing resources unused at times and thus do not optimally leverage them yet. We present look-ahead scheduling, a wall-time minimizing parallelization strategy for ABC Sequential Monte Carlo algorithms, which avoids idle times of computing units by preemptive sampling of subsequent generations. This allows to utilize all available resources. The strategy can be integrated with e.g. adaptive distance function and summary statistic selection schemes, which is essential in practice. Our key contribution is the theoretical assessment of the strategy of preemptive sampling and the proof of unbiasedness. Complementary, we provide an implementation and evaluate the strategy on different problems and numbers of parallel cores, showing speed-ups of typically 10-20% and up to 50% compared to the best established approach, with some variability. Thus, the proposed strategy allows to improve the cost and run-time efficiency of ABC methods on high-performance infrastructure.

## Introduction

Mechanistic models are important tools in systems biology and many other research fields to describe and understand mechanisms underlying systemic behavior [1, 2]. Usually, such models have unknown parameters that need to be estimated by comparing model outputs to observed data [3]. For complex stochastic models, in particular multi-scale models used to describe the complex dynamics of multi-cellular systems, evaluating the likelihood of data given parameters however becomes quickly computationally infeasible [4, 5]. For this reason, simulation-based methods that circumvent likelihood evaluation have been developed, such as

code and data can be found at https://doi.org/10.5281/zenodo.7875905.

**Funding:** The authors acknowledge the Gauss Centre for Supercomputing e.V. (www.gauss-centre.eu) for funding this project by providing computing time on the GCS Supercomputer JUWELS at Jülich Supercomputing Centre (JSC). This work was supported by the German Federal Ministry of Education and Research (BMBF) (FitMultiCell/031L0159C and EMUNE/031L0293C) and the German Research Foundation (DFG) under Germany's Excellence Strategy (EXC 2047 390873048 and EXC 2151 390685813 and the Schlegel Professorship for JH). YS acknowledges support by the Joachim Herz Foundation. FG was supported by the Chica and Heinz Schaller Foundation. There was no additional external funding received for this study.

**Competing interests:** The authors have declared that no competing interests exist.

approximate Bayesian computation (ABC), popular for its simplicity and wide applicability [6, 7].

ABC generates samples from an approximation to the true Bayesian posterior distribution. While asymptotically exact, a known disadvantage of ABC is its computational complexity. The reason for this is that it requires often simulations of hundred thousands to millions of artificial datasets. Therefore, methods to efficiently explore the search space have been developed [8]. In particular, ABC is frequently combined with a Sequential Monte Carlo scheme (ABC-SMC), which over several generations successively refines the posterior approximation via importance sampling while maintaining high acceptance rates [9, 10]. Furthermore, in ABC-SMC the sampling for each generation can be parallelized, enabling the use of high-performance computing (HPC) infrastructure. This has in recent years enabled tackling increasingly complex problems via ABC [11–14].

It would be desirable if available computational resources were perfectly exploited at all times, to minimize both the wall-time until results become available to the researcher, and the cost associated with allocated resources. However, the problem is that established parallelization strategies to distribute ABC-SMC work over a set of workers leave resources idle at times and thus fall short of this aim. The parallelization strategy used in most established HPC-ready ABC implementations is *static scheduling (STAT)*, which defines exactly as many parallel tasks as accepted particles are required [15, 16]. While it minimizes the active compute time and consumed energy, typically a substantial amount of workers become idle towards the end of each generation. *Dynamic scheduling (DYN)* mitigates this problem and reduces the overall wall-time by continuing sampling on all workers until sufficiently many particles have been accepted [17]. It was shown to reduce the wall-time substantially. However, also in this strategy at the end of each generation workers become idle, waiting until all simulations have finished.

A natural strategy to circumvent idle time is to start already sampling the next generation, given partial information about the current generation. Yet, it is not obvious how particles need to be accepted or weighted, and whether this would indeed improve efficiency. In this manuscript, we describe an ABC-SMC parallelization strategy for multi-core and distributed systems, called *look-ahead scheduling (LA)* which avoids idle time. We show that by appropriate sample reweighting we obtain an unbiased Monte Carlo sample. We provide an HPC-ready implementation and test the method on various problems. Moreover, we show that the strategy can be integrated with adaptive algorithms for e.g. summary statistics, distance functions, or acceptance thresholds.

## Methods

### ABC

We consider a mechanistic model described via a generative process of simulating data $y \sim \pi(y|\theta) \in \mathbb{R}^{n_y}$ for parameters $\theta \in \mathbb{R}^{n_\theta}$. Given observed data $y_{\text{obs}}$, in Bayesian inference the likelihood $\pi(y_{\text{obs}}|\theta)$ is combined with prior information $\pi(\theta)$ to the posterior distribution $\pi(\theta|y_{\text{obs}}) \propto \pi(y_{\text{obs}}|\theta) \cdot \pi(\theta)$. We assume that evaluating the likelihood is computationally infeasible, but that it is possible to simulate data $y \sim \pi(y|\theta)$ from the model. Then, classical ABC consists in the 3 steps of first sampling parameters $\theta \sim \pi(\theta)$, second simulating data $y \sim \pi(y|\theta)$, and third accepting $\theta$ if $d(y, y_{\text{obs}}) \leq \varepsilon$, for a distance metric $d : \mathbb{R}^{n_y} \times \mathbb{R}^{n_y} \to \mathbb{R}_{\geq 0}$ and acceptance threshold $\varepsilon > 0$. This is repeated until sufficiently many particles, say $N$, are accepted. The population

of accepted particles constitutes a sample from an approximation of the posterior distribution,

$$\pi_{\mathrm{ABC},\varepsilon}(\theta|y_{\mathrm{obs}}) \propto \int I[d(y, y_{\mathrm{obs}}) \leq \varepsilon] \pi(y|\theta) dy \cdot \pi(\theta). \qquad (1)$$

Under mild assumptions, $\pi_{\mathrm{ABC},\varepsilon}(\theta|y_{\mathrm{obs}})$ converges to the actual posterior $\pi(\theta|y_{\mathrm{obs}})$ as $\varepsilon \to 0$ [18, 19]. Commonly, ABC operates not directly on the measured data, but summary statistics thereof, capturing relevant information in a low-dimensional representation [20]. Here, for notational simplicity we assume that $y$ already incorporates summary statistics, if applicable.

## ABC-SMC

The vanilla ABC formulation exhibits a trade-off between the reduction of the approximation error induced by $\varepsilon$, and high acceptance rates. Thus, ABC is frequently combined with a Sequential Monte Carlo scheme (ABC-SMC) [21, 22]. In ABC-SMC, a series of particle populations $P_t = \{(\theta_t^i, w_t^i)\}_{i \leq N}$ is generated, constituting samples of successively better approximations $\pi_{\mathrm{ABC},\varepsilon_t}(\theta|y_{\mathrm{obs}})$ of the posterior, for generations $t = 1, \ldots, n_t$, with acceptance thresholds $\varepsilon_t > \varepsilon_{t+1}$. In the first generation ($t = 1$), particles are sampled directly from the prior, $g_1(\theta) = \pi(\theta)$. In later generations ($t > 1$), particles are sampled from proposal distributions $g_t(\theta) \gg \pi(\theta)$ based on the last generation's accepted weighted population $P_{t-1}$, e.g. via a kernel density estimate. The importance weights $w_t^i$ are the Radon-Nikodym derivatives $w_t(\theta) = \pi(\theta)/g_t(\theta)$. This is precisely such that the weighted parameters are samples from the distribution

$$\int w_t(\theta) I[d(y, y_{\mathrm{obs}}) \leq \varepsilon_t] \pi(y|\theta) dy \cdot g_t(\theta) = \int I[d(y, y_{\mathrm{obs}}) \leq \varepsilon_t] \pi(y|\theta) dy \cdot \pi(\theta), \qquad (2)$$

i.e. the target distribution (1) for $\varepsilon = \varepsilon_t$.

Common proposal distributions first select an accepted parameter from the last generation and then perturb it, in which case $g_t$ takes the form $g_t(\theta) = \sum_{i=1}^{N} w_{t-1}^i K(\theta|\theta_{t-1}^i) / \sum_{i=1}^{N} w_{t-1}^i$, with e.g. $K(\theta|\theta_{t-1}^i) = \mathcal{N}(\theta|\theta_{t-1}^i, \Sigma_{t-1})$ a normal distribution with mean $\theta_{t-1}^i$ and covariance matrix $\Sigma_{t-1}$. The performance of ABC-SMC algorithms relies heavily on the quality of the proposal distribution, on its ability to efficiently explore the parameter space. Methods that adapt to the problem structure, e.g. basing $\Sigma_{t-1}$ on the previous generation's weighted sample covariance matrix and potentially localizing around $\theta_i$, have shown superior performance [23–25].

The output of ABC-SMC is a population of weighted parameters

$$P_{n_t} = \{(\theta_{n_t}^i, w_{n_t}^i)\}_{i \leq N} \sim \pi_{\mathrm{ABC},\varepsilon_{n_t}}(\theta|y_{\mathrm{obs}}).$$

For a statistic $f : \mathbb{R}^{n_\theta} \to \mathbb{R}$, the expected value under the posterior is then approximated via the self-normalized importance estimator

$$\mathbb{E}_{\pi_{\mathrm{ABC},\varepsilon_{n_t}}(\theta|y_{\mathrm{obs}})}[f] \approx \hat{f} = \sum_{i=1}^{N} W_{n_t}^i f(\theta_{n_t}^i),$$

which is asymptotically unbiased. Here, $W_t^i := w_t^i / \sum_{j=1}^{N} w_t^j$ are self-normalized weights. This is necessary because the weights $w_t(\theta) = \pi(\theta)/g_t(\theta)$ are not normalized in the joint sample space $(\theta, y)$, therefore effectively another Monte Carlo estimator is employed for the normalization constant (for details see the Section 1.1 in S1 File).

In importance sampling, samples are assigned different weights, such that some impact estimates more than others. This can be quantified e.g. via the *effective sample size (ESS)* [8, 26]:

$$\text{ESS}(\{w_i\}_{i \leq N}) = \frac{\left(\sum_{i \leq N} w_i\right)^2}{\sum_{i \leq N} w_i^2} \qquad (3)$$

## Established parallelization strategies

In ABC, often hundred thousands to millions of model simulations need to be performed, which is typically the computationally critical part. To speed up inference, parallelization strategies have been developed that exploit the independence of the *N* particles constituting the *t*-th population. Suppose we have *W* parallel workers, each worker being a computational processor unit e.g. in an HPC environment. There are two established techniques to parallelize execution over the workers:

In *static scheduling (STAT)*, given a population size *N*, *N* tasks are defined and distributed over the workers. Each task consists in sampling until one particle gets accepted (Fig 1A). The tasks are queued if *N* > *W*. STAT minimizes the active computation time and number of simulations and is easy to implement, only requiring basic pooling routines available in most distributed computation frameworks. However, even for *W* > *N* only *N* workers are employed, although the number of required simulations is usually substantially larger than *N*. In addition, at the end of every generation the number of active workers decreases successively, most workers idly waiting for a few to finish their tasks. STAT is available in most established ABC-SMC implementations [15, 16].

In *dynamic scheduling (DYN)*, sampling is performed continuously on all available workers until *N* particles have been accepted (Fig 1B). However, simply taking those first *N* particles as the final population would bias the population towards parameters causing short-running simulations. Therefore, DYN waits for all workers to finish, and out of then $\tilde{N} \geq N$ accepted particles, only the *N* that started earliest are finally accepted, not the ones that finished earliest. This has the effect that simulation time plays no longer a role in the acceptance decision [17]. This ensures that the acceptance probability of a particle is in accordance with the target distribution, independent of later events and thus its run-time.

## Parallelization using look-ahead dynamic scheduling

DYN allows to exploit the available parallel infrastructure to a higher degree than STAT and therefore already substantially decreases the wall-time (by a factor of between 1.4 and 5.3 in test scenarios, see [17]). Nonetheless, some workers remain idle at the end of each generation while waiting for others to complete. This fraction of idle workers increases as the number of workers increases relatively to the population size. Additionally, the idle time increases if simulation times are heterogeneous, which is often the case, e.g. with estimated reaction rates determining the number of simulated events (Section 3.4.1 in S1 File). In case of fast model simulations, also the time between generations, e.g. to post-process and store results, may be relatively long.

**Proposed algorithm.** We propose to extend dynamic scheduling by using the free workers at the end of each generation to proactively sample for the next generation (Fig 2): As soon as *N* acceptances have been reached in generation *t* − 1 and workers thus start to get idle, we construct a preliminary proposal $\tilde{g}_t$, based on which particles for generation *t* are generated to start simulations on the free workers. $\tilde{g}_t$ can be based on a preliminary population of accepted

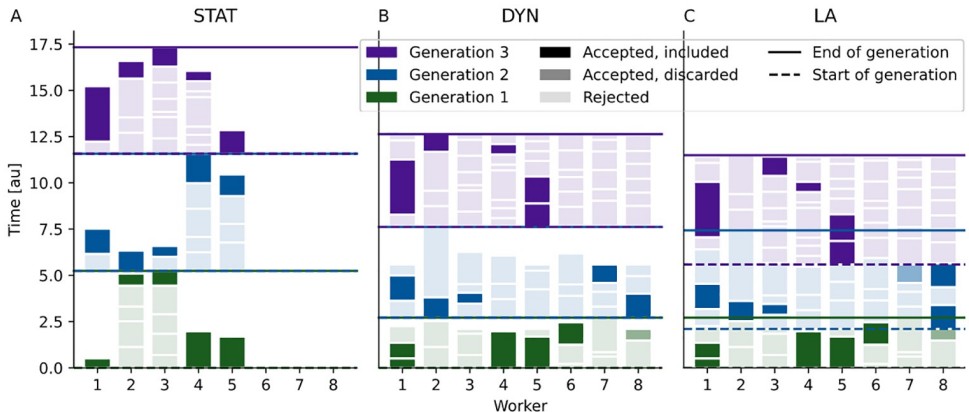

**Fig 1. Illustration of core usage over run-time for static (STA), dynamic (DYN) and look-ahead (LA) scheduling for a population size $N = 5$ on $W = 8$ workers, over 3 generations (colors).** The shading associated with each color indicates whether a sample satisfies the acceptance criterion and is included in the final population (dark shading), satisfies the acceptance criterion but is discarded because enough earlier-started accepted samples exist (intermediate shading, for DYN+LA), or does not satisfy the acceptance criterion and is rejected (light shading). Solid lines indicate the end of a generation, dashed lines indicate the (preliminary) beginning of a generation (different from solid lines only for LA). Delimiting white spaces between boxes indicate negligible post-processing times between simulations.

particles $\hat{P}_{t-1} = \{(\hat{\theta}_{t-1}^i, \hat{w}_{t-1}^i)\}_{i \leq N}$ relying on these first $N$ acceptances. However, $\hat{P}_{t-1}$ may introduce a practical bias (in a finite sample sense) towards particles with faster simulations times. This can in particular occur when computation time is highly parameter-dependent. Say, for example, that parameters from multiple regions in parameter space can explain the data similarly well, but that one region leads to substantially higher simulation times. Then, a sampling routine that does not wait for all started simulations to finish may under-represent or even miss out on regions in parameter space with high simulation times. The ABC-SMC routine may consequently have a low probability of generating importance samples from that region in subsequent generations, leading to a biased final posterior sample. To address this issue, the preliminary proposal can alternatively be based on $P_{t-2}$ (such that $\tilde{g}_t = g_{t-1}$), giving

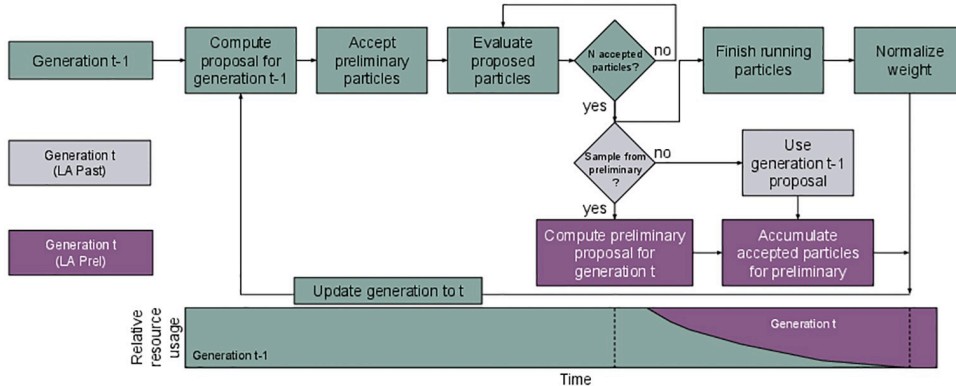

**Fig 2. Concept visualization of look-ahead scheduling (LA).** As soon as no more simulations are required for generation $t - 1$ (green), a preliminary simulation task for generation $t$ is formulated either based on population $P_{t-2}$ (grey, LA Past) or the preliminary population $P_{t-1}$ (purple, LA Prel). Resulting simulations are considered when evaluating the next generation, and suitable weight normalization is applied to all samples (top right). Over time, the number of workers dedicated to generation $t - 1$ decreases, while that for generation $t$ increases (bottom).

inductively practically unbiased proposals. If a particle $\tilde{\theta}_t \sim \tilde{g}_t$ gets accepted according to the acceptance criteria of generation $t$, its non-normalized weight is calculated as $\tilde{w}_t(\tilde{\theta}_t) = \frac{\pi(\tilde{\theta}_t)}{\tilde{g}_t(\tilde{\theta}_t)}$. As soon as all simulations for generation $t - 1$ have finished and thus the actual $P_{t-1}$ is available, all workers are updated to continue working with the actual sampling task based on proposal $g_t$. As the time-critical part of typical ABC applications is the model simulation, the cost of generating the preliminary sampling task is usually negligible.

The assessment of acceptance of preliminary samples depends on whether everything is pre-defined: If the acceptance components, including distance function $d$ and acceptance threshold $\varepsilon_t$ for generation $t$ are defined a-priori, then acceptance can be checked directly on the workers without knowledge of the complete previous population $P_{t-1}$. If however any component of the algorithm is adaptive and hence based on $P_{t-1}$ (e.g. the acceptance threshold is commonly chosen as a quantile of $\{d(y_{t-1}^i, y_{\mathrm{obs}})\}_{i \leq N}$), the acceptance check must be delayed until the actual $P_{t-1}$ is available. This allows to use one common acceptance criterion across all particles within a generation, so that all particles target the same distribution.

The population of generation $t$ is then, corrected for run-time bias as in DYN by only considering the $N$ accepted particles that were started first, given as

$$P_t = \{\{(\tilde{\theta}_t^i, \tilde{w}_t^i)\}_{i \leq \tilde{N}}, \{(\theta_t^i, w_t^i)\}_{\tilde{N} < i \leq N}\}, \tag{4}$$

with $0 \leq \tilde{N} \leq N$ particles based on the preliminary proposal $\tilde{g}_t$, and $N - \tilde{N}$ on the final $g_t$. The weights need to be normalized appropriately, as explained in the following section. We call this parallelization strategy, which during generation $t - 1$ already looks ahead to generation $t$, *Look-ahead (dynamic) scheduling (LA)*.

**Weights and unbiasedness.** A key property of ABC methods is that they provide an asymptotically unbiased Monte Carlo sample from $\pi_{\mathrm{ABC},\varepsilon_{n_t}}(\theta|y_{\mathrm{obs}})$, with $\pi_{\mathrm{ABC},\varepsilon_{n_t}}(\theta|y_{\mathrm{obs}}) \to \pi(\theta|y_{\mathrm{obs}})$ for $\varepsilon \to 0$. The sample (4) obtained via LA conserves this property: The point is that each subpopulation on its own gives an asymptotically unbiased estimator, since the weights $\tilde{w}_t(\tilde{\theta}) = \pi(\tilde{\theta})/\tilde{g}_t(\tilde{\theta})$, $w_t(\theta) = \pi(\theta)/g_t(\theta)$ are exactly the Radon-Nikodym derivatives w.r.t. the respective proposal distributions. Note that this theoretical unbiasedness holds regardless of whether $\tilde{g}_t$ is based on $\hat{P}_{t-1}$ or $P_{t-2}$, as long as $\tilde{g}_t(\theta) \gg \pi(\theta)$ As noted in the previous section, however a practical bias may occur due to finite sample size.

The subpopulation estimates are then combined, which decreases the Monte Carlo error due to the larger sample size. Instead of simply tossing all samples together, it is preferable to first normalize the weights relative to their subpopulation, $\tilde{W}_t^i := \tilde{w}_t^i / \sum_{i=1}^{\tilde{N}} \tilde{w}_t^j$, $W_t^i := w_t^i / \sum_{i=\tilde{N}+1}^{N} w_t^i$ (Section 1.3 in [S1 File]). This is because both weight functions are non-normalized, with generally different normalization constants, which renders them not directly comparable. A joint estimate based on the full population can then be given as

$$\mathbb{E}_{\pi_{\mathrm{ABC},\varepsilon_t}(\theta|y_{\mathrm{obs}})}[f] \approx \beta \sum_{i=1}^{\tilde{N}} \tilde{W}_t^i f(\tilde{\theta}_t^i) + (1 - \beta) \sum_{i=\tilde{N}+1}^{N} W_t^i f(\theta_t^i) \tag{5}$$

with $\beta \in [0, 1]$ a free parameter. A straightforward choice is $\beta = \tilde{N}/N$, rendering the contribution of each subpopulation proportional to the respective number of samples. Instead, we propose to choose $\beta$ to maximize the overall effective sample size (3), rendering the Monte Carlo estimate more robust. This is a simple constrained optimization problem with solution

$$\beta = \frac{\mathrm{ESS}(\{\tilde{W}_t^i\}_{i \leq \tilde{N}})}{\mathrm{ESS}(\{\tilde{W}_t^i\}_{i \leq \tilde{N}}) + \mathrm{ESS}(\{W_t^i\}_{\tilde{N} < i \leq N})}$$

i.e. the contribution of each subpopulation is proportional to its effective sample size (Section 1.4 S1 File). Supposing that for $N \to \infty$, $\tilde{N}/N \to \alpha \in [0, 1]$, (5) converges to the left-hand side, as required. A more detailed derivation and extension to more than two proposal distributions is given in the Section 1 in S1 File.

## Implementation and availability

We implemented LA in the open-source Python tool pyABC [27], which already provided STAT and DYN. We employ a Redis low-latency server to handle the task distribution. If all components are pre-defined, we perform evaluation of the "look-ahead" samples $(\tilde{\theta}, \tilde{y})$ directly on the workers. If there are adaptive components, the delayed evaluation is performed on the main process. To avoid generating unnecessarily many preliminary samples in the presence of some very long-running simulations, we limited the number of preliminary samples to a default value of 10 times the number of samples in the current iteration. To not start preliminary sampling unnecessarily, we employed schemes predicting whether any termination criterion will be hit after the current generation. The code underlying this study can be found at https://github.com/EmadAlamoudi/Lookahead_study. A snapshot of code and data can be found at https://doi.org/10.5281/zenodo.7875905.

## Results

Wall-time superiority of DYN over STAT has already been established in prior work [17]. To study the performance of LA and compare it to DYN, we applied both to several parameter estimation problems and in various scenarios of population size $N$ and workers $W$. We distinguish between "LA Prel" using the preliminary $\hat{P}_{t-1}$ to generate $\tilde{g}_t$, and "LA Past" using $P_{t-2}$ instead.

## Test problems

We considered four problems (Table 1): Problems T1-T2 are simple test problems, while M1-M2 are realistic application examples.

Problem T1 is a bimodal model $y \approx \theta^2$, in which simulations from one mode have an artificially longer run-time. Specifically, if $\theta > 0$, a log-normally distributed simulation time of $\tau \sim \log\mathcal{N}(1, 2, 4)$ seconds was simulated. The goal of setting an artificially longer run-time was to specifically test the preliminary bias caused by only accepting simulations from one mode when constructing the preliminary proposal.

Problem T2 is an ordinary differential equation (ODE) model with 2 parameters describing a conversion reaction $x_1 \leftrightarrow x_2$, with observables obscured by random multiplicative noise. To analyze sampler behavior under simulation run-time heterogeneity, we added random log-normally distributed delay times $t_{\text{sleep}}$ of various variances on top of the ODE simulations. For this model, run-times are fast, permitting repeated analyses to check correctness of the method, quantify stochastic effects and assess average behavior.

**Table 1. Overview of application examples.**

| ID | Description | Implementation | $n_\theta$ |
|---|---|---|---|
| T1 | Bimodal run-time-skewed model | Python | 1 |
| T2 | Conversion reaction ODE model | Python | 2 |
| M1 | Tumor spheroid growth [11] | C++ | 7 |
| M2 | Liver tissue regeneration [28] | Morpheus | 14 |

Problem M1 describes the growth of a tumor spheroid using a hybrid discrete-continuous approach, modeling single cells as stochastically interacting agents and extracellular substances [11]. The model combines a system of PDEs to describe the extracellular matrix, with a cellular Potts model (CPM) to describe cell configurations and mechanisms like cell division and cell death. The model has seven estimated parameters and outputs three observables, the growth curve, extra-cellular matrix and proliferation profiles.

Problem M2 describes the metabolic status of mechano-sensing during liver generation, describing the reaction network dynamics by a set of ODEs [28]. This model has 14 parameters and two observables, the nuclear YAP and total YAP intensities. These observables were quantified from image tiles covering an entire liver lobule with portal and central veins.

Further details about the test problems can be found in the Section 3 in S1 File.

## Biased proposal can induce practical bias in accepted population

The analysis of test model T1 revealed that for small population sizes $N$ relative to the number of workers $W$, in combination with high acceptance rates, LA Prel can indeed lead to a bias towards short-running simulations (Fig 3 right). This can happen when $\hat{P}_{t-1}$ is only based on short-running simulations, and solely proposes particles from that regime, enough of which are then accepted to form $P_t$. For larger $N$ relative to $W$, this effect occurred less, likely because given large population sizes, sampling from other modes with associated high importance weights eventually happened.

## Sampling from unbiased proposal solves bias

When replacing LA Prel by LA Past, i.e. sampling from $P_{t-2}$ instead of $\hat{P}_{t-1}$, the bias no longer occurred (Fig 3 right). This is expected, because $\tilde{g}_{t-2}$ has no run-time bias. In practice, we did not encounter any problems of practical bias on the considered application examples, where results from DYN, LA Prel and LA Past were highly consistent. Yet, LA Prel may fail in some situations, which also demonstrates that ABC-SMC algorithms are sensitive to potential bias in the proposal distribution. Thus, in the following, we focus on the stable LA Past algorithm, showing pendants for LA Prel in the S1 File.

## Look-ahead sampling gives accurate results

We used problem T2 to analyze different scenarios, with population sizes $N$ from 20 to 1280 particles, worker numbers $W$ from 32 to 256, and log-normally distributed simulation times of variances $\sigma^2$ from 1.0 to 4.0. We ran each scenario 13 times to obtain stable average statistics. We considered means and standard deviations as point and uncertainty measures.

Point estimates for DYN and LA converged to the same values across population sizes (Fig 4A and 4B). The proportion of accepted LA samples in the final population originating from the preliminary distribution ranged from nearly 100% to 50% (LA Prel) and 20% (LA Past), as expected decreasing for larger population sizes (Fig 4E and 4F). The more pronounced decrease for LA Past than LA Prel is reasonable because void of bias, $\hat{P}_{t-1}$ provides a better sampling distribution than $P_{t-2}$. Effective sample sizes were stable across DYN and LA (Fig 4D). A higher run-time variance lead to an increase in accepted samples originating from the preliminary proposal distribution (S1 Fig in S1 File). This is expected, because greater heterogeneity in run-times increases the chance of encountering exceptionally long-running simulations, which DYN has to wait for, while LA already proceeds.

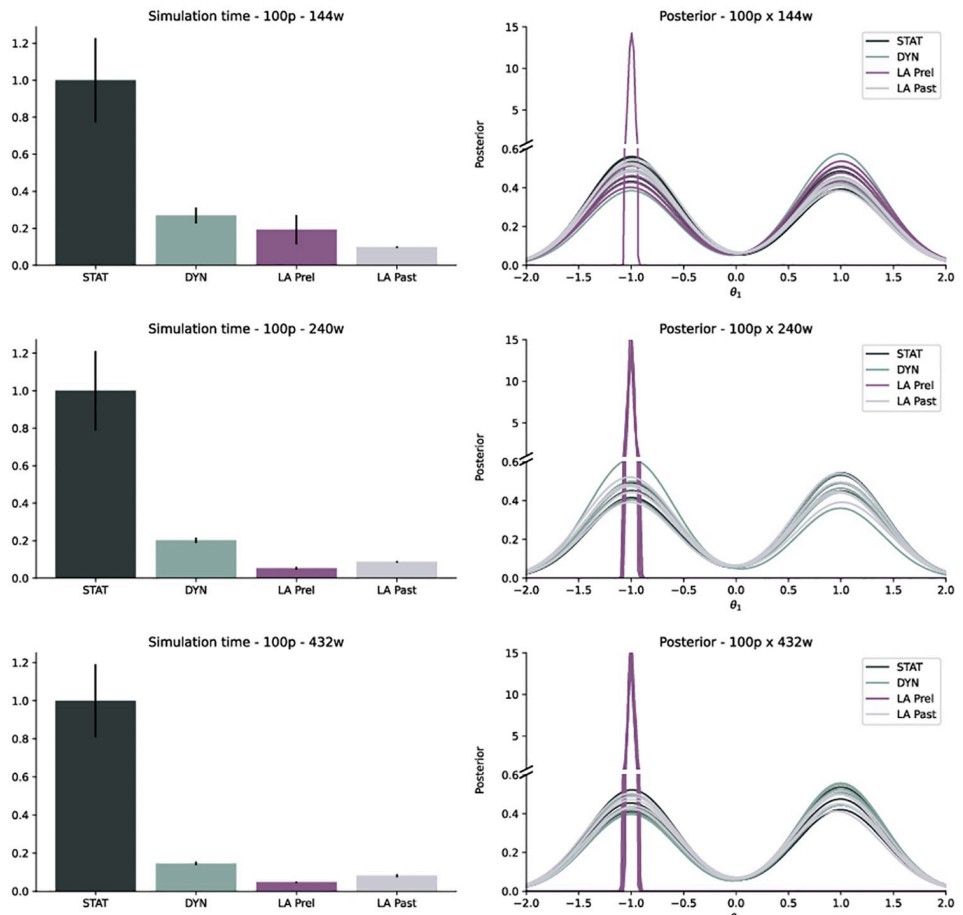

**Fig 3. Run-time and posterior approximation for 5 different runs of model T1 with STAT, DYN, LA Prel and LA Past, with population size $N = 100$ on $W = 144, 240, 432$ workers (top to bottom).**

### Considerable speed-up towards high worker numbers

To analyze the effect of scheduling strategy on the overall wall-time, we ran model T2 systematically for different population sizes and numbers of workers. We considered population sizes $32 \leq N \leq 1280$ and numbers of parallel workers $32 \leq W \leq 256$, which covers typical ranges used in practice. Each scenario was repeated between 13 times to assess average behavior, here we report mean values.

As a general tendency, the wall-time speed-up of LA over DYN became larger with increasing ratio of the number of workers by the population size. For a model sleep time variance of $\sigma^2 = 1$ (Fig 5), e.g. for $N = 20$ and $W = 256$, the average wall-time got reduced by a factor of almost 1.8. In most scenarios, a wall-time reduction by a factor of between 1.11 and 1.8 was observed. Only when the population size was large compared to the number of workers, the speed-up was comparably small. Generally, the increase in speed-up with increasing ratio $W/N$ is as expected, as for large $W$ the idle time between generations occurring for DYN and LA constitutes a more pronounced factor in the overall run-time.

For a sleep time variance of $\sigma^2 = 2$ (S2 Fig in S1 File), we observed similar behavior. There, the acceleration was generally more pronounced with up to a factor of roughly 1.9 and many

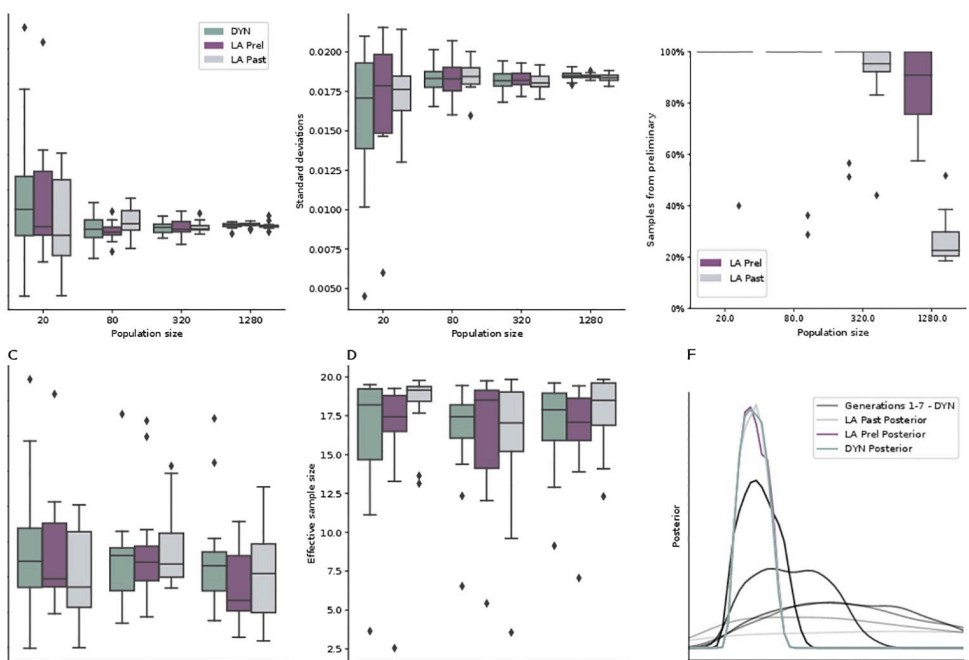

**Fig 4. Results for problem T2 for different population sizes *N*, worker numbers *W*, and sleep time variances $\sigma^2$.**
Unless otherwise specified, we used $N = 1280$, $W = 256$, and a log-normally distributed sleep time $t_{\text{sleep}}$ of variance $\sigma^2 = 1$. To increase comparability, the $\varepsilon_t$ values over $n_t = 8$ generations were pre-defined. (A) and (B): Mean and standard deviation of the posterior approximation $\pi_{\text{ABC},\varepsilon_{n_t}}(\theta|y_{\text{obs}})$. Box-plot over 13 repetitions. (C): Posterior mean for different sleep time variances, for $W = 20$. (D): Effective sample size across different sleep time variances, for $N = 256$ and $W = 20$, in which case it is likely that several generations are sampled completely from the preliminary proposal. (E): Fraction $\tilde{N}/N$ of accepted samples in the final population $t = n_t$ that originate from the preliminary proposal $\tilde{g}_{n_t}(\theta)$ for LA Prel and LA Past. (F): Exemplary visualization of 1d posterior approximation marginals for single runs.

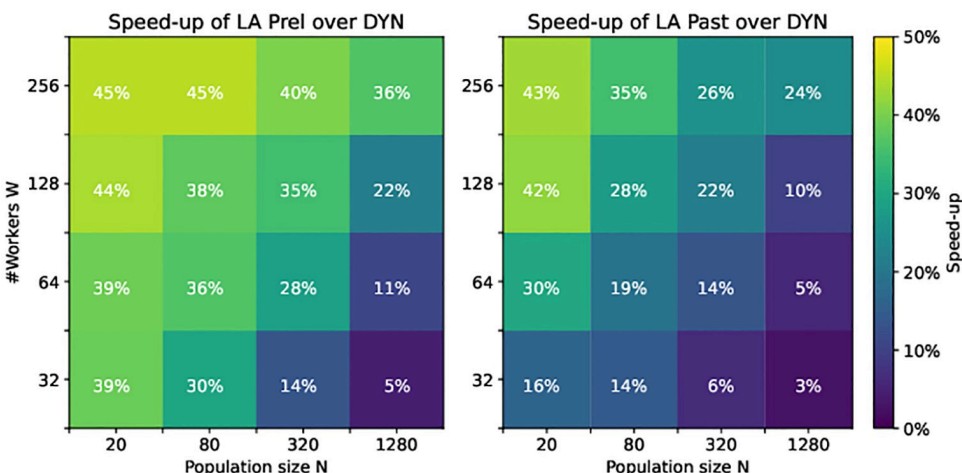

**Fig 5. Speed-up (1 − {Wall- time LA}/{Wall- time DYN}) of LA Prel (left) and LA Past (right) over DYN for various population sizes and numbers of workers, for a model sleep time variance of $\sigma^2 = 1$.**

factors in the range 1.2 to 1.9. This indicates that indeed the advantage of LA over DYN is more pronounced in the presence of highly heterogeneous model simulation times.

Also on T1, the comparison of run-times (Fig 3 left) revealed a speed-up of LA over DYN. Further, we could confirm on both T1 and T2 (Fig 3 and S3 Fig in S1 File) the substantial speed-up DYN already provides over STAT, as reported in [17], on which we here improved further.

## Scales to realistic application problems

Given the high simulation cost of the application problems M1–2, we only performed selected analyses to compare LA and DYN. A reliable comparison of run-times in real-life application examples is challenging, because the total wall-time varies strongly due to stochastic effects, and computations are too expensive to perform inference many times.

For the two models, the parameter estimates obtained using LA (both LA Per and LA Past) and DYN are consistent, except for expectable stochastic effects (S4 and S5 and S9–S11 Figs in S1 File). Together with the previous analyses, this indicates that for practical applications, the multi-proposal approach of LA allows for stable and accurate inference, similar to the single proposal used by DYN. In early generations, a considerable part of the accepted particles was based on the preliminary proposal distribution (near 100%), which then decreased in later generations (S6 and S12 Figs in S1 File). This is consistent with the decrease in acceptance rate and thus the relative time during which the preliminary and not the final proposal distribution is used.

For the tumor model M1, we used an adaptive quantile-based epsilon threshold schedule [29], with DYN, LA Prel and LA Past, population sizes $N \in \{250, 500, 1000\}$, and $W \in \{128, 256\}$ workers. For each considered configuration we performed 2 replicates (in total 8) to assess average behavior. Reported run-times are until a common final threshold was hit by all runs. The speed-up of LA over DYN varied depending on the ratio of population size and number of workers, similar to what we observed for T1+2. For high ratios, LA was consistently faster up to 35%. However, for low ratios, less improvement was observed. In some runs, LA was slightly slower than DYN (Fig 6). Over the 8 runs, we observed a mean speed-up of 21% (13%) and a median of 23% (16%) for LA Past (LA Prel). This indicates expected speed-ups of 13–23%. However, it should be remarked that large run-time differences and volatility could be traced back to single generations taking vast amounts of time (S7 Fig in S1 File). These long generations occurred in all scheduling variants and exist most likely because the epsilon for that generation was chosen too optimistically, indicating a weakness of the used epsilon scheme rather than the parallelization strategy.

For the liver regeneration model M2, we performed similar analyses, with adaptive quantile-based epsilon threshold schedules, population sizes $N \in \{250, 500, 1000\}$ and $W \in \{128, 256\}$ workers, with 2 replicates per configuration. Similar to model M1, we observed a faster performance of up to 35%. However, with a smaller ratio between population size and the number of workers, a slightly lower performance improvement was achieved (Fig 7). Similarly to M1, the acceleration varied quite strongly. For LA Prel we observed a mean speed-up over all 8 runs of 36% (median 31%) over DYN. However, for LA Past we observed contrarily a mean slow-down of 39% (median 43%) over DYN. It is not clear what caused this stark difference, which is again subject to high fluctuations. Further tests would be needed to assess the reasons for this specific model.

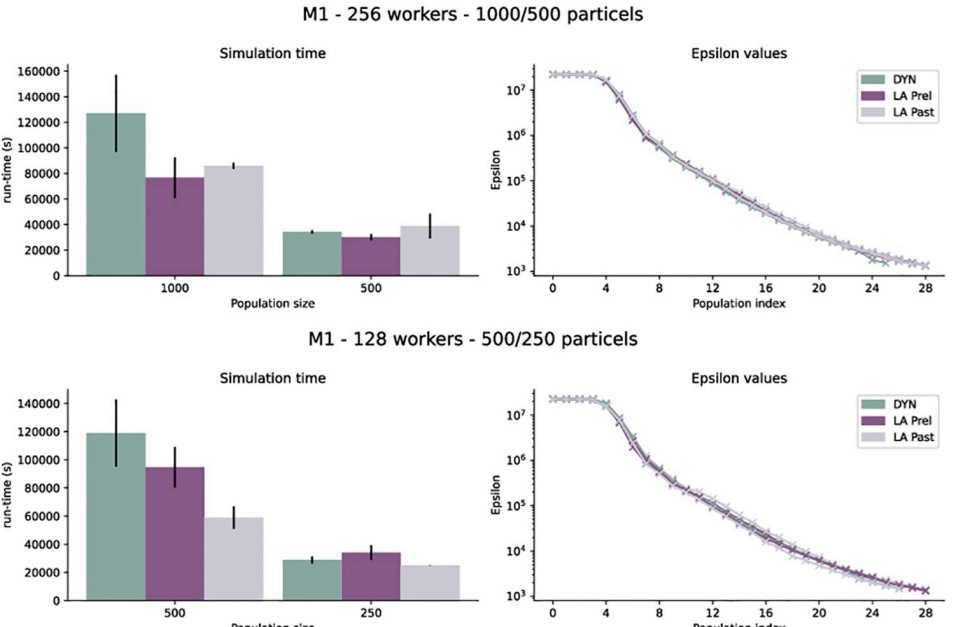

**Fig 6. Run-time and posterior distributions for 2 different runs of model M1 with population size 1000, 500, 250 on 128 and 256 workers.**

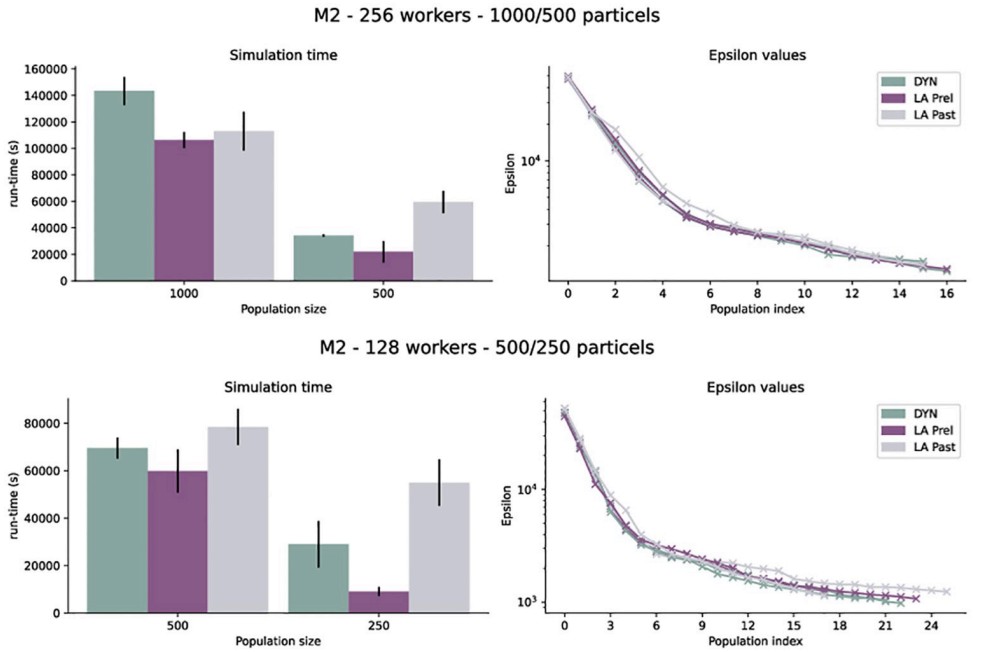

**Fig 7. Run-time and posterior distribution for 2 different runs of model M2 with population size 1000, 500, 250 on 128 and 256 workers.**

## Discussion

Simulation-based ABC methods have made parameter inference increasingly accessible even for complex stochastic models, which are however limited by computational costs. Here, we presented "look-ahead" sampling, a parallelization strategy to minimize wall-time and improve run-time efficiency by using all available high-performance computing resources at near-all times. On various test and application examples, we verified the accuracy and robustness of the novel approach in typical settings. Depending on model simulation run-time heterogeneity, and the relation of population size and the number of available cores, we observed a speed-up of up to 45% compared to dynamical scheduling as the previously most efficient strategy. Compared to widely used static scheduling, dynamic scheduling is already highly efficient, with limited room for improvement. Nevertheless, using the here proposed look-ahead sampling, on typical application examples, we observed a speed-up of often roughly 20–30%, however with some variability and sometimes efficiency on par with or even below dynamical scheduling. Assessing these variations in efficiency in more detail on expensive application examples would require further tests with considerable computational resources. Importantly, our analysis also demonstrates how ABC-SMC is sensitive to the choice of proposal distribution. Finite samples can induce a practical bias, as we observed here for parameter-dependent run-times of models—a problem that occurred in extreme cases but could only be solved by using look-ahead sampling with the previous, and not the preliminary, proposal distribution.

Conceptually and aside implementation details, the presented strategy provides the minimal wall-time among all parallelization strategies, as all cores are used at practically all times. We observed that look-ahead sampling using preliminary results (LA Prel) provided a performance speed-up over re-using the previous generation (LA Past), however at the cost of practical bias. Thus, LA Past constitutes the safe choice. Only if the possibility of critical parameter-dependent simulation times can be excluded, would we presently recommend LA Prel.

Were it possible to construct an unbiased proposal using those preliminary results, e.g. via reweighting or imbalance detection, we could thus increase the speed-up with robust performance. Alternatively, LA Past and LA Prel could also be combined, e.g. switching to LA Prel after a "burn-in", when the probability of a bias toward short-running simulations is lessened.

When using delayed evaluation, it would be possible to parallelize the evaluation as well, which we have not done here. If evaluation times are long relative to simulation times, e.g. if (adaptive) summary statistics involve complex operations, this would be beneficial. In order to reduce a potential bias in the preliminary proposal distribution towards fast-running simulations, it may be beneficial to update it as soon as more particles finish. This would imply the use of more than two importance distributions, the theory of which we have already provided in the S1 File.

In conclusion, we showed how we can minimize wall-time and associated computing cost of ABC samplers with substantial performance gains over established methods. Given that the concept is generally applicable for sequential importance sampling methods, it is of potential widespread use for different applications.

## Supporting information

**S1 File. Further details on the methods and results.**
(PDF)

## Author Contributions

**Conceptualization:** Jan Hasenauer, Yannik Schälte.

**Formal analysis:** Yannik Schälte.

**Funding acquisition:** Jan Hasenauer.

**Investigation:** Emad Alamoudi, Felipe Reck, Yannik Schälte.

**Methodology:** Emad Alamoudi, Felipe Reck, Yannik Schälte.

**Software:** Emad Alamoudi, Felipe Reck, Yannik Schälte.

**Supervision:** Jan Hasenauer, Yannik Schälte.

**Validation:** Emad Alamoudi, Felipe Reck, Nils Bundgaard, Frederik Graw, Lutz Brusch.

**Visualization:** Emad Alamoudi, Felipe Reck.

**Writing – original draft:** Emad Alamoudi, Felipe Reck, Jan Hasenauer, Yannik Schälte.

**Writing – review & editing:** Emad Alamoudi, Jan Hasenauer, Yannik Schälte.

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
