## [Decision Letter · Decision Letter 0]

3 Sep 2023

PONE-D-23-20726A Wall-time Minimizing Parallelization Strategy for Approximate Bayesian ComputationPLOS ONE

Dear Dr. Schälte,

Thank you for submitting your manuscript to PLOS ONE. After careful consideration, we feel that it has merit but does not fully meet PLOS ONE’s publication criteria as it currently stands. Therefore, we invite you to submit a revised version of the manuscript that addresses the points raised during the review process.

We look forward to receiving your revised manuscript.

Kind regards,

Abel C.H. Chen

Academic Editor

PLOS ONE

Journal Requirements:

"The authors acknowledge the Gauss Centre for Supercomputing e.V. (www.gauss-centre.eu) for

funding this project by providing computing time on the GCS Supercomputer JUWELS at Jülich

Supercomputing Centre (JSC). This work was supported by the German Federal Ministry of Edu-

cation and Research (BMBF) (FitMultiCell/031L0159C and EMUNE/031L0293C) and the German

Research Foundation (DFG) under Germany’s Excellence Strategy (EXC 2047 390873048 and EXC

2151 390685813 and the Schlegel Professorship for JH). YS acknowledges support by the Joachim

Herz Foundation. FG was supported by the Chica and Heinz Schaller Foundation."

Reviewers' comments:

Reviewer's Responses to Questions

**Comments to the Author**

1. Is the manuscript technically sound, and do the data support the conclusions?

Reviewer #1: Yes

Reviewer #2: Yes

2. Has the statistical analysis been performed appropriately and rigorously? 

Reviewer #1: Yes

Reviewer #2: Yes

3. Have the authors made all data underlying the findings in their manuscript fully available?

Reviewer #1: Yes

Reviewer #2: Yes

4. Is the manuscript presented in an intelligible fashion and written in standard English?

Reviewer #1: Yes

Reviewer #2: Yes

5. Review Comments to the Author

Reviewer #1: The article "A Wall-time Minimizing Parallelization Strategy for Approximate Bayesian Computation" proposed a Look-ahead scheduling technique to fully utilize the idle time that address issues the current static and dynamic scheduling have. Two kinds of LA scheme, LA pre and LA cur are proposed and compared through numerous numerical examples. t is clearly written and will be a useful contribution to the community. link to the software is provided. I was able to access the code through the link provided in the paper. A few minor comments for the manuscript that will be useful for the authors to discuss but does not require any further calculations for this paper, are as below.

1. In section 2.4.1, could the authors add more discussion about the bias introduced by LA Pre, such as, where does the bias come from, why particles with faster simulation times tend to introduce a practical bias?

2. In Figure 3, the authors illustrate the bias caused by LA pre scheduling method. However, it's hard for me to tell the information from the right hand-side plot. I was wondering if a more detailed explanation for the problematic realization (the purple "vertical" lines), or updated plots with extended y-axis can be posted?

3. In section 3.5, what's the reason behind the slower performance as the ratio between number of population size and number of workers increase?

4. How do you determine which method (LA pre and LA cur) is the better choice given a problem?

Reviewer #2: In this study, E. Alamoudi and his colleagues develop a novel parallelization strategy implemented in an high-performance computing (HPC) infrastructure. This strategy aims at improving the speed, mainly by reducing wall-time, of Approximate Bayesian Computation (ABC) sequential Monte Carlo algorithm. The authors show that their new strategy is unbiased and converges to the expected value of Monte Carlo sampling. They also compare their strategy (look-ahead scheduling) with existing parallelization strategies, assessing relative performance at four different test problems. Given its flexibility, ABC is a popular approach that can be applied to very complex models. As such, any potential speed-up strategy is useful and could prove to be valuable for many researchers. Thus, it is my recommendation that this manuscript should be accepted. Nevertheless, I have a few comments and suggestions. Please note that, although I have worked with ABC before, I am not a computational scientist and thus all my comments will be aimed at improving the clarity of the manuscript for a more general audience. I hope my comments are useful and contribute to make this submission more valuable for PlosOne wide readership.

- I think the authors limit the scope of the manuscript by starting the introduction with systems biology. Although “systems biology” is an umbrella term that encompasses several fields and research questions, this parallelization strategy could be useful in other fields such as population genomics or human health, for instance. I think a more general start is warranted, possibly by just mentioning the variety of applications of ABC, including systems biology.

- “While asymptotically exact, a known disadvantage of ABC is its reliance on repeated simulation, often hundred thousands to millions of times” - while this is technically correct, further ahead the authors clearly state the 3 steps of classical ABC. Thus, I think this could be rephrased to highlight that a disadvantage of ABC is the computationally heavy and time consuming simulation of data points (or datasets), which is usually the most time consuming step of ABC.

- I believe there is an unfinished sentence in the “ABC-SMC” section of the Methods. The sentence that starts with “Methods that adapt to the problem structure” ends with “have shown superior”. I think a word is missing after “superior”.

- Although Figure 1 is useful to understand the comparison between the different parallelization strategies, I found it somewhat confusing. Particularly, it was confusing to see shades of gray in the Figure key but not in the Figure itself. I think the authors could remove the shading from the Figure key and include that information only in the Figure legend itself. Possibly as: “The shading associated with each color indicates whether a sample satisfies the acceptance criterion and is included in the final population (dark shading), satisfies the acceptance criterion but is discarded because enough earlier-started accepted samples exist (intermediate shading, for DYN+LA), or does not satisfy the acceptance criterion and is rejected (light shading).” Additionally, it is not clear what the blank areas in-between shaded areas represent for each worker? Idle times or post-process?

- Throughout the text the authors refer to Figure 1A, 1B and 1C but I did not see any A, B or C indicated in Figure 1.

- When explaining static scheduling (STAT), the authors mention that the “tasks are queued if N ≥ W”. Do tasks really get queued if N = W? It was my understanding that if N workers are available then all tasks will start and no task will be queued.

- “Therefore, DYN waits for all workers to finish, and out of then N ≥ N accepted particles, only the N that started earliest are finally accepted”. Is this correct? This sentence is confusing, because if there is a bias towards accepting particles that end earlier, how would waiting for all workers to finish and then selecting the N that started earlier solve that bias?

- This might just be a personal preference but I don’t understand the rationale behind the naming convention chosen for “LA Cur” and “LA Pre”. Although I understand that “LA Pre” refers to the use of a Preliminary sample, “LA Cur” makes me think of “Current” which is less than intuitive given that it is using an earlier generation Pt-2. Maybe this will be clearer for other readers but, for me, it would make more sense to call these two approaches “LA Pre” for preliminary and “LA Pge” for the past generation.

- In the subsection “Test problems”, I think a few more general details about the problems considered would make the results more understandable to a wider audience. I suggest adding a table with the parameters for each problem to the supplementary information. The authors should also mention that a detailed explanation of the test problems can be found in the supplementary information. Additionally, I think the authors should expand their explanation of T1 since this problem is important to understand the preliminary bias associated with “LA Pre”. Particularly, I think that they should more clearly state that the aim of the problem is to infer a posterior distribution with two modes (i.e., a bimodal distribution), which is a tricky problem. Furthermore, it could be made clearer that the goal of setting an artificially longer run-time was to specifically test the preliminary bias caused by only accepting simulations from one mode when constructing gt based on Pt-1.

- In the section titled “Sampling from unbiased proposal solves bias” the authors correctly state that “LA Pre may fail in some situations, which also demonstrates that ABC-SMC algorithms are sensitive to potential bias in the proposal distribution”. Given sufficient number of generations, is it possible to use “LA Cur” in the first generations, as a sort of “burn-in”, and then switch to “LA Pre” when the probability of a bias towards short-running simulations is lessened?

- In section “3.6 Scales to realistic application problems”, is it possible that the “mean slow-down of 39% (median 43%) over DYN” observed for “LA Cur” is a result of a slower refinement of the preliminary proposal distribution? In other words, for a sufficiently complex model, basing the proposal distribution on Pt−2 implies that the search space for each generation is more vast than it would be if proposal distribution was based on Pt−1. Could this explain the observed slow-down?

6. PLOS authors have the option to publish the peer review history of their article (what does this mean?). If published, this will include your full peer review and any attached files.

Reviewer #1: No

Reviewer #2: No

---

## [Author Response · Author response to Decision Letter 0]

13 Oct 2023

Please find our response to all reviewer comments in the Response PDF file.

---

## [Decision Letter · Decision Letter 1]

25 Oct 2023

A Wall-time Minimizing Parallelization Strategy for Approximate Bayesian Computation

PONE-D-23-20726R1

Dear Dr. Schälte,

We’re pleased to inform you that your manuscript has been judged scientifically suitable for publication and will be formally accepted for publication once it meets all outstanding technical requirements.

Kind regards,

Abel C.H. Chen

Academic Editor

PLOS ONE

Additional Editor Comments (optional):

Reviewers' comments:

Reviewer's Responses to Questions

**Comments to the Author**

1. If the authors have adequately addressed your comments raised in a previous round of review and you feel that this manuscript is now acceptable for publication, you may indicate that here to bypass the “Comments to the Author” section, enter your conflict of interest statement in the “Confidential to Editor” section, and submit your "Accept" recommendation.

Reviewer #1: All comments have been addressed

Reviewer #2: All comments have been addressed

2. Is the manuscript technically sound, and do the data support the conclusions?

Reviewer #1: Yes

Reviewer #2: Yes

3. Has the statistical analysis been performed appropriately and rigorously? 

Reviewer #1: Yes

Reviewer #2: Yes

4. Have the authors made all data underlying the findings in their manuscript fully available?

Reviewer #1: Yes

Reviewer #2: Yes

5. Is the manuscript presented in an intelligible fashion and written in standard English?

Reviewer #1: Yes

Reviewer #2: Yes

6. Review Comments to the Author

Reviewer #1: (No Response)

Reviewer #2: After carefully re-reading the updated manuscript and the authors replies, I feel that the authors addressed all comments of the two reviewers, both in their response letter and in the updated manuscript. I appreciate the change in nomenclature and the detailed description of the test problems. I believe that this has improved the manuscript, which is now clearer and more comprehensive.

7. PLOS authors have the option to publish the peer review history of their article (what does this mean?). If published, this will include your full peer review and any attached files.

Reviewer #1: No

Reviewer #2: No

---

## [Editor Report · Acceptance letter]

3 Nov 2023

PONE-D-23-20726R1 

A Wall-time Minimizing Parallelization Strategy for Approximate Bayesian Computation 

Dear Dr. Schälte:

I'm pleased to inform you that your manuscript has been deemed suitable for publication in PLOS ONE. Congratulations! Your manuscript is now with our production department. 

Kind regards, 

on behalf of

Dr. Abel C.H. Chen 

Academic Editor

PLOS ONE